# Automated Detection of Change of Direction in Basketball Players Using Xsens Motion Tracking

**DOI:** 10.3390/s25030942

**Published:** 2025-02-05

**Authors:** Salvatore Pinelli, Raffaele Zinno, Anna Jòdar-Portas, Anna Prats-Puig, Raquel Font-Lladó, Laura Bragonzoni

**Affiliations:** 1Department for Life Quality Studies, University of Bologna, Corso D’Augusto 237, 47921 Rimini, Italy; salvatore.pinelli2@unibo.it (S.P.); laura.bragonzoni4@unibo.it (L.B.); 2University School of Health and Sport (EUSES), University of Girona, 17190 Salt, Spain; ajodar@euses.cat (A.J.-P.); aprats@euses.cat (A.P.-P.); rfont@euses.cat (R.F.-L.); 3Research Group Health and Health Care, Nursing Department, University of Girona, 17071 Girona, Spain; 4Research Group of Culture and Education, Institute of Educational Research, University of Girona, 17007 Girona, Spain; 5Chair of Sport and Physical Education-Center of Olimpic Studies, University of Girona, 17820 Banyoles, Spain

**Keywords:** sport science, agility test, V-cut, COD, activity segmentation, wearable sensors

## Abstract

In sports science, accurate tracking of athletes’ movement patterns is essential for performance analysis and injury prevention. Changes of direction (COD), frequently executed during basketball games at cutting angles of around 135° (internal angle of 45°), are essential for agility and high-level performance. Moreover, mastering effective COD mechanics is associated with a lower risk of injuries and enhanced long-term athletic success. However, manual segmentation of data from wearable sensors is labor-intensive and time-consuming, often creating bottlenecks for sports practitioners. The aim of this study was to evaluate the feasibility and accuracy of an automated algorithm for detecting COD movements in basketball and to compare its performance with manual detection methods. Data were collected from 62 basketball players, each completing two tests (V-cut test and a modified V-cut test), totaling 248 trials. The system utilizes kinematic data from an Xsens full-body kit to analyze key variables that characterize direction changes. The proposed method detects COD events with a median error of one frame and an interquartile range of two frames. The system demonstrated nearly 80% accuracy in COD detection, as validated against manual video analysis. These findings indicate that automated COD detection can significantly reduce segmentation time for practitioners while providing actionable, data-driven insights to enhance kinematic assessment during sport-specific activities.

## 1. Introduction

Wearable sensors have become increasingly significant in sports science for capturing kinematic and motion-related information in real-time, particularly within the field environment. This technology provides an efficient alternative to traditional lab-based systems, enabling researchers to analyze movements in realistic sport-specific contexts [1]. Indeed, wearable sensor technology enables the assessment of key aspects of sports activities, such as speed and acceleration, providing valuable insights into complex abilities such as agility [1,2,3].

Agility is a fundamental component of most team sports and is defined as the ability to move quickly by altering speed or direction in response to an external stimulus [4]. From a biomechanical perspective, a change of direction (COD) involves a deceleration phase, consisting of several braking steps, followed by a propulsive phase directed towards the intended direction [5]. A key event in every COD is the final foot contact (FFC), defined as the last step before the player transitions to the new direction. Equally important is the step preceding the FFC, particularly during the sharp turns. Such a step, known as the penultimate foot contact (PFC), is crucial for braking prior to the COD [6]. For CODs exceeding 60°, it is recommended to apply significant braking force during the PFC [7], which may be dangerous for the joints. The eccentric–concentric muscular activity required for these movements may increase the likelihood of stress or muscle damage in players [5,8,9].

Basketball players perform up to 835 directional changes during a game [10], with an average sprinting distance of approximately 20 m and each high-to-moderate intensity effort lasting 2–3 s in adolescent players [11]. The most common cutting angle during match play is approximately 135° (±15°), measured relative to the initial running direction (equivalently, a 45° internal angle) [12], often referred to as a V-cut or side-step cutting maneuver [13]. Most existing tests for evaluating COD ability in team sport athletes involve movement patterns that may lack sport-specific relevance [14,15]. The V-cut test was developed to better reflect the cutting actions commonly observed in basketball players [16]. Efficiency in COD mechanics has been associated with greater agility and is considered an indicator of superior performance in elite athletes [17]. Moreover, effective COD mechanics may lower the risk of long-term injuries [5,18]. Consequently, evaluating the quality of these movements and quantifying them over time could provide valuable insights for enhancing player performance [19].

COD kinematic data analysis often relies on manual segmentation, which, although precise, is time-consuming and labor-intensive, limiting its practicality for large-scale applications. This process requires specialized expertise to ensure accuracy and consistency, increasing costs and resource demands. As data acquisition rates increase in real-time and high-volume studies, the scalability of manual methods becomes a significant constraint [20], especially in environments requiring rapid data turnaround, such as professional sports. Moreover, human error is an inherent risk in manual methods. Fatigue and cognitive overload can lead to mistakes, reducing overall data quality and diminishing the reliability of research findings [21,22].

For kinematic and inertial measurement unit (IMU) data, automated approaches that eliminate the need for extensive manual input provide a scalable solution for high-throughput environments. In motion analysis, automated segmentation has similarly shown promise in improving precision and reliability for IMU-tracked gestures [23]. Kratz and Back [24] demonstrated the ability of automated systems to perform complex segmentation tasks with a high level of fidelity using a machine learning classifier. Moreover, advances in deep learning have further pushed the boundaries of automated segmentation in IMU data processing [25]. This growing body of research affirms the substantial potential of automated segmentation methods, which not only streamline data processing, but also enable scalable solutions for real-time and high-throughput applications.

Several sports adopt automatic segmentation and event identification in kinematic data. In running, for example, foot contact events are commonly detected using maxima/minima of kinematic variables [26,27]. However, these methods may not capture the specific phases of basketball COD movements. A minimum trunk velocity or a fixed time window around it does not necessarily align with key foot contact events in basketball, nor does it adequately account for the biomechanical demands of sudden deceleration.

Although these methods are well-documented in other sports, they have not been integrated for basketball COD at the 135° angle. By adapting foot contact detection approaches from running [26,27] and combining them with horizontal velocity analysis [5,28], a segmentation strategy is introduced to address the unique demands of sudden decelerations in basketball. This helps fill a gap in the existing literature, where the standard methods used for linear activities often do not fully capture the complexity of basketball maneuvers.

In light of this, the aim of this study was to evaluate the feasibility and accuracy of an automated algorithm for detecting COD movements in basketball and to compare its performance with manual detection methods.

We hypothesized that the automated approach, which uses specific kinematic parameters, would demonstrate accuracy comparable to that of manual methods, while providing a more efficient and reproducible analysis.

## 2. Materials and Methods

This is a methodological comparative study that evaluated COD identification during the V-cut test, comparing the performance of an automated algorithm (heuristic rule-based) with that of a manual tracking method. We conducted an experiment to train and test the algorithm, comparing its estimation results with those of the manual approach. First, we described the experimental setup, the phases of the COD identified, and data processing procedures. Then, we explained the automated method for comparison. Finally, we described the procedure for training and testing the proposed method.

### 2.1. Design

This research was developed in youth teams of a professional basketball club from Girona (Spain). The Research Ethics Committee of the Dr. Josep Trueta hospital in Girona approved the research (ID: 2020.193), which conformed to the recommendations of the Declaration of Helsinki. Informed consent and assent were obtained from all subjects and their parents.

### 2.2. Participants

The sample included 62 (33 males and 29 females) young basketball players from 13 to 18 years of age recruited from the Bàsquet Girona club (Table 1). All players carried out 3 ninety-minute training sessions and a game per week. The inclusion criteria required active involvement in basketball training, either at a competitive or recreational level, no history of lower-limb injuries or surgeries within the past six months, and a willingness to participate in all testing sessions. The exclusion criteria included any current injuries or conditions that could affect performance during the agility tests.

### 2.3. Experimental Setup

The V-cut test was conducted under two different conditions: without ball (standard V-cut test) and with ball (modified V-cutBk test). This dual approach aimed to evaluate the algorithm’s performance in both general athletic and sport-specific tasks, such as dribbling while handling a ball in order to closely simulate real match conditions. The V-cut test consisted of a 25 m sprint with four CODs (two on the right leg and two on the left leg). Cones were arranged in a “V” shape, ensuring players crossed a marked line at each turn (to anticipate sidestep cutting movements) and requiring an actual COD at a 135° angle with respect to the athlete’s original running direction (corresponding to an internal angle of 45°). Each segment of the run measured 5 m in length (Figure 1).

The V-cutBk test design was based on the V-cut test protocol [16,29]. The V-cutBk test included dribbling, with subjects starting while holding the ball with both hands and performing crossover dribbles during CODs. The V-cut test design involves changes of direction to both the right and left sides, ensuring that performance evaluation is independent of participants’ leg dominance. The complete experimental procedures, including anthropometric measurements and familiarization sessions, were described in a previous study [29]. The tests were conducted on the same indoor basketball court, and subjects adhered to specific pre-testing guidelines, such as avoiding stimulants, maintaining consistent nutritional habits, and refraining from vigorous exercise 24 h before the sessions.

### 2.4. Change of Direction Phases

The COD movement was divided into two main phases: the eccentric phase (braking) and the concentric phase (pushing). The transition between the two phases occurs approximately when the trunk velocity reaches the minimum value [30,31]. Two distinct events were identified around this minimum to account for the braking and pushing phases: the initial frame (IF) and the final frame (FF). The IF is defined as the heel strike of the PFC of the non-cutting foot. Instead, FF is defined as the toe off of the FFC of the cutting foot (Figure 2 and Appendix A).

### 2.5. Data Collection

The MVN Biomech Link system (Xsens Technologies BV, Enschede, The Netherlands), a previously validated setup comprising 17 inertial measurement units, a transmission pack, and a battery [32], was used for the kinematic assessment. Calibration required participants to assume the N-pose, characterized by standing upright with their upper limbs relaxed and parallel to the body, lower limbs fully extended, and feet positioned parallel [33]. Anthropometric measurements were incorporated to scale the proprietary biomechanical model within the Xsens Motion Tracking System, facilitating motion capture calibration via Xsens MVN Analyze software (version 2023.2). The Xsens MVN system has demonstrated fair to excellent accuracy in sagittal plane motion analysis when compared to gold-standard optoelectronic systems, making it well suited for capturing sport-specific, on-field movements [32]. During testing, the participants wore their own basketball shoes to ensure comfort and natural movement patterns. Each participant performed the V-cut test and the V-cutBK test twice (four trials in total), with an interval of approximately 30 s between each trial.

### 2.6. Data Processing

A total of 248 tests (124 V-cut and 124 V-cutBk) were collected and were used for the comparison analysis. The kinematics collected on the field were extracted from the Xsens MVN Analyze 2023.2 software suite (Xsens Technologies, Enschede, The Netherlands) and further processed in a customized script in Python (Python Software Foundation; Python Language Reference, version 3.10; Python Software Foundation: Wilmington, DE, USA).

#### 2.6.1. Manual Segmentation

For the validation of the algorithm, it is important to label each of the defined segments from the recorded data. The ground truth (manual segmentation) refers to the manually annotated video analysis data, where the annotator identified the IF and FF of the four CODs of the V-cut test. The manual segmentation dataset was created by two internship students in the field of sport science. Each segmentation was independently reviewed by two authors (S.P. and R.Z.). In cases of disagreement on the identified frame, the authors discussed their evaluations to reach a consensus. This process ensured a comprehensive assessment of each frame while reducing the likelihood of misdetections. This newly created outcome variable acted as the gold standard, which was used to evaluate the algorithm’s accuracy.

#### 2.6.2. Automatic Segmentation

The automatic segmentation of COD was performed using an automated algorithm (heuristic rule-based) that, starting from the MVNX data, outputs the frames corresponding to the beginning (IF) and end (FF) of the COD. The algorithm’s design incorporated methodologies derived from the existing literature on kinematic analysis, with adaptations to suit the specific context of COD movements in basketball.

The process begins with MVNX data, including segment and joint kinematics and foot contact information. The trunk’s horizontal velocity (Vhor) is calculated and then low-pass filtered to reduce high-frequency noise. Next, the local minima of Vhor are identified. Around each minimum, a search region is defined to detect the IF and FF based on changes in foot contact. These algorithm-derived IF and FF can then be compared with manually segmented frames. A workflow diagram illustrating the full process, from data acquisition to result validation, is shown in Figure 3.

Foot contact data and trunk velocity were extracted from the MVNX file. The “foot contact” is an internal feature of the Xsens MVN Analyze 2023.2 software suite. By analyzing the full-body kinematics, this tool identifies whether a foot is in contact with the ground at each time frame, assigning a value of 1 for contact and 0 for no contact [26].

The trunk’s horizontal velocity (V_hor_) was computed by combining its velocity components along the horizontal plane (x and y axes). Then, the V_hor_ was filtered with a fourth-order two-way Butterworth low-pass filter with a cut-off frequency of 1.5 Hz [34,35]. The filtered data were analyzed to identify the local minima, shown in Figure 4 with black dots.

For each identified minimum, a search region was established. Within the search region, transitions between contact and non-contact states in the foot contact columns were analyzed to identify the initial frame (IF) and final frame (FF) (Appendix A). Specifically, the IF was defined as the frame where the foot contact value of the PFC was 1 and its first derivative was also 1. Similarly, the FF was defined as the frame where the foot contact value of the FFC was 0 and its first derivative was 1 (Figure 4).

### 2.7. Model Training and Test

To ensure reliable performance and minimize the risk of overfitting, the dataset was divided into distinct training and test sets, following an 80:20 split. Gender, team categories, and class types were balanced across the training and test sets (Appendix A). The training set, consisting of 796 CODs, was used to develop the algorithm by fine-tuning its parameters, such as the prominence of local velocity minima and the amplitude of the search region for IF and FF identification. Various prominence values and search-region amplitudes were tested, and the combination that yielded the lowest root mean square error (RMSE) between automatic and manual segmentation was a prominence of 1.5 and a search region of −40 to +25 frames around these minima. These optimized parameters were then applied to the test set, comprising 196 CODs, which was reserved for evaluating the model’s performance on previously unseen data.

### 2.8. Statistical Analysis

FF and IF obtained from the automated segmentation were compared to the FF and IF from the manual segmentation of the 4 different CODs of the V-cut and V-cutBk tests. We report the absolute mean error |E| and the standard deviation of the error s(E). The average precision (AP) for different frame thresholds was computed. For instance, AP_1_ represents the proportion of correctly identified instants (in this case, with an absolute error equal or below 1 frame) relative to the total number of observations N [36].

For the test dataset, Bland–Altman plots were created for both conditions separately [37]. Additionally, the Pearson correlation coefficient and coefficient of determination (R^2^) between total cutting time for each COD were used to assess the relationship between manual and automated segmentation, providing insight into the reliability of the automated segmentation performance metrics [38,39].

## 3. Results

Differences between methods (computed as automatic minus manual segmentation), analyzed across various tests, showed a median difference of 1 frame with an interquartile range (IQR) of approximately 2–3 frames (Table 2).

The average |E| and its s(E) varied slightly between tests in training dataset, with |E| ranging from 1.2 to 1.8 and s(E) being approximately equal to 2 frames. In the test dataset, the performance metrics show a slight decrease compared to the training dataset. Nevertheless, the results remain optimal, indicating that the algorithm generalizes effectively under the given experimental conditions. In this dataset, the |E| values range from 1.7 to 2.9, while s(E) ranges from 2.7 to 4.6. The initial frame (IF) detection generally achieved higher average precision (AP) than final frame (FF) detection in the training dataset at more stringent accuracy thresholds (Table 3). For example, in the V-cut in the training dataset, IF reached AP3 values near 94.7% compared to 88.6% for FF. However, the test dataset results were more mixed, with IF occasionally surpassing FF. For instance, in V-cut in test dataset, IF recorded 89.6% at AP3, while FF reached 92.7%.

The Bland–Altman plots did not show a systematic trend for the estimation errors. The limits of agreement (LoAs) for the Bland–Altman plots of the Vcut (Figure 5) and VcutBK (Figure 6) tests showed differences in the agreement between IF and FF. In both tests, IF values exhibited larger LoAs compared to FF.

Figure 7 shows moderate to high correlations between the manually and automatically determined total cutting times for both V-cut and V-cut Bk conditions. In the V-cut trials, the coefficient of determination (R^2^ = 0.50) and Pearson’s correlation (r = 0.74) suggest a relatively strong association between the two methods. Similarly, in the V-cut Bk condition (R^2^ = 0.39, r = 0.68), the relationship remains clear, though slightly weaker.

## 4. Discussion

The adoption of an automated algorithm significantly accelerates the analysis of large datasets compared to traditional manual segmentation methods. Although manual segmentation is precise, it is time-consuming, prone to human error, and relies heavily on trained professionals, leading to bottlenecks in time-sensitive scenarios. As the demand for real-time data processing and large-scale studies grows, the limitations of manual methods (inefficiency and vulnerability to human error) underscore the need for scalable and reliable alternatives.

This study addresses these challenges by proposing an automated method, which achieved promising accuracy, reducing segmentation errors to a median difference of 33 ms (2 frames) and an IQR of approximately 50 ms (2–3 frames) across all tests. These results underscore the efficiency and reliability of the algorithm in identifying COD events, even under varying test conditions. These findings are consistent with prior research that highlights the advantages of automated segmentation methods in motion analysis [23,40]. Kratz and Back (2015) [24] achieved a precision of 0.95 in identifying various phases of gesture execution using a machine learning classifier, capturing initiation, transition, and completion phases with minimal need for manual intervention. This success demonstrates the ability of automated systems to perform complex segmentation tasks with a high level of fidelity, addressing the limitations of manual methods while maintaining accuracy and reproducibility. Apte et al. [23] developed an automated method for detecting the CODs on agility *t*-tests based on the wavelet-denoised antero-posterior acceleration signal, which was developed and validated using video data (60 Hz) that effectively reduced the subjective nature of manual segmentation with a relative error for COD duration less than 3.5 ± 16% [23]. Moreover, a combined convolutional neural network and recurrent network model developed by Zimmermann et al. [25] achieved an accuracy of 98.57% in assigning IMUs to specific body segments, with a median angular error of just 2.91° for alignment tasks. This growing body of research affirms the substantial potential of automated segmentation methods, which not only streamline data processing but also enable scalable solutions for real-time and high-throughput applications.

The R^2^ and Pearson correlation coefficient values for the relationship between manually and automatically determined total cutting time were moderate. Although these metrics indicate a fairly strong association, they appear to be influenced by outliers in the dataset.

These outliers, resulting from occasional misdetection, contributed to a slightly lower-than-expected correlation. Despite its overall accuracy, the automated method occasionally wrongly detected foot contact events. For instance, Figure 8 illustrates mislabeled frames during COD events, highlighting specific scenarios where the algorithm encountered challenges.

The choice of the local minimum as the reference point can, in certain cases, lead to classification errors. Indeed, an error in detecting the IF is presented, where the manually identified frame falls outside the defined search region (Figure 8a). Another error is observed in identifying the FFC (Figure 8b). In this case, the local minimum detected by the algorithm occurred after the actual FF, and as a result, the algorithm failed to capture the true toe off of the FFC, instead identifying the subsequent one.

In both cases of misdetection, the common factor appears to be the determination of the local minimum. In the first case, the local minimum was identified too far forward (Figure 8a), causing the interval to shift to the right. In the other case, the local minimum exceeded the actual FF value (Figure 8b), leading to incorrect detection. These examples underline how the reliance on the local minimum as a reference point can sometimes result in errors, particularly in scenarios where the motion is irregular, or sensor misalignments occur.

These misdetections suggest that, while the automated algorithm generally performs well, certain conditions (irregular movement patterns or slight misalignments in the sensor data) introduce variability that affects the consistency and accuracy of the results. These findings highlight areas where the algorithm could be refined to improve its robustness and reliability in detecting critical events, extending its utility in complex athletic scenarios and reinforcing its value in sports science applications.

Overall, the V-cut test showed promising applications in both its standard and modified forms in athletic performance monitoring, injury prevention, and talent identification. By tracking COD mechanics over time, practitioners can identify agility deficits or movement inefficiencies that may increase an athlete’s injury risk. Additionally, the test’s sensitivity to training-related improvements makes it a useful tool for monitoring athletic development throughout a competitive season. Its demonstrated applicability to adolescent basketball players suggests potential use in talent scouting, particularly for identifying athletes with exceptional agility and COD efficiency, both key indicators of elite performance in basketball and other team sports [16].

Despite the algorithm demonstrating strong accuracy and reliability, some limitations should be addressed. First, the presence of outliers, as discussed previously, suggests variability in certain conditions or among specific participants. Improving preprocessing steps (such as outlier detection and correction) through refining by incorporating additional kinematic parameters to the algorithm to better differentiate foot contact states, or applying machine learning techniques to detect more subtle transitions, could help mitigate this issue.

Second, the algorithm’s applicability to other sports or movement patterns has yet to be tested, limiting its generalizability for now. Third, the algorithm currently relies on the MVNX file format of the Xsens motion system, particularly for its direct provision of foot contact events (e.g., toe off and heel strike). This limits the algorithm’s direct applicability to other motion capture systems. However, the algorithm could be adapted by importing motion data from Excel/CSV files and utilizing alternative methods to identify foot contact and trunk’s horizontal velocity events through kinematic analyses (e.g., sacrum accelerations [41] or foot accelerations [27]). These adaptations would enable broader applicability while preserving the algorithm’s fundamental structure and capabilities. Future research could focus on several areas: integrating real-time feedback systems to enhance the practical application of the test, adapting the algorithm for other sport-specific movements to broaden its usability, and conducting longitudinal studies to evaluate the V-cut test’s ability to predict injury risks or performance outcomes over time.

## 5. Conclusions

The automated COD detection system proposed in this study, based on trunk horizontal velocity and foot contact extracted from MVNX files, demonstrated high accuracy and precision in identifying COD events during a V-cut test, closely matching the performance of manual segmentation. This segmentation enables the identification of multiple COD movements at a 135° angle, allowing for further kinematic analysis focused on this critical time period. This method holds significant potential as an effective tool for COD performance evaluation.

## Figures and Tables

**Figure 1 sensors-25-00942-f001:**
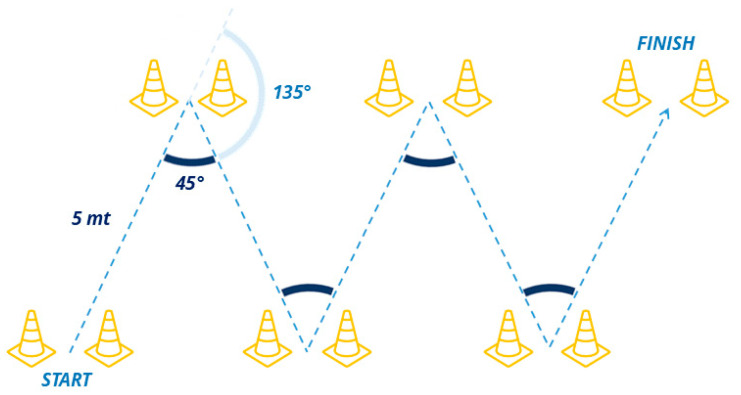
Schematic illustration of V-cut test. The dashed blue line represents the path followed by the athlete, while the dashed light-blue lines indicate the intended running direction before each change of direction.

**Figure 2 sensors-25-00942-f002:**
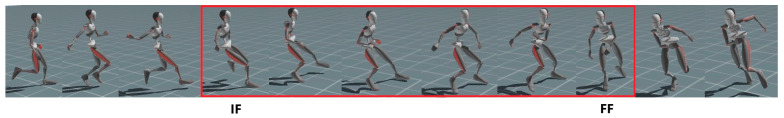
Schematic illustration of IF and FF identification. The red box indicates the phases in which the subject transitions from IF to FF.

**Figure 3 sensors-25-00942-f003:**
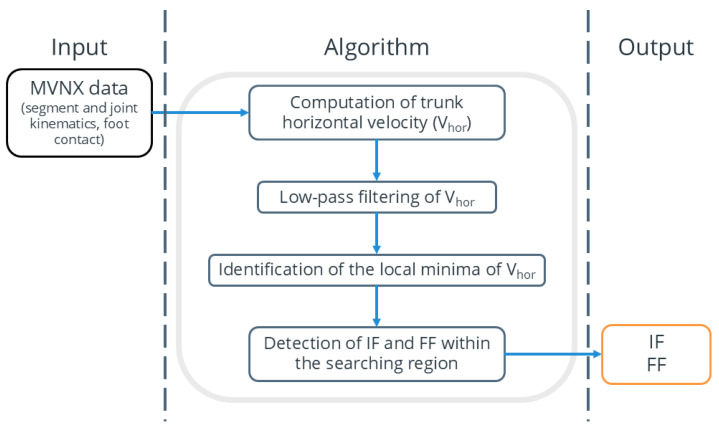
Algorithm workflow for extracting the initial frame (IF) and final frame (FF) from an MVNX file.

**Figure 4 sensors-25-00942-f004:**
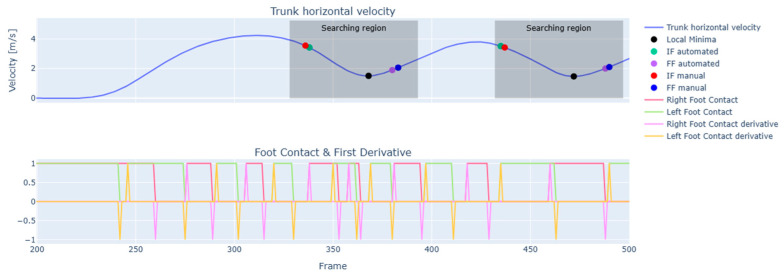
Schematic illustration of FF and IF identification from horizontal trunk velocity and foot contact and its first derivative.

**Figure 5 sensors-25-00942-f005:**
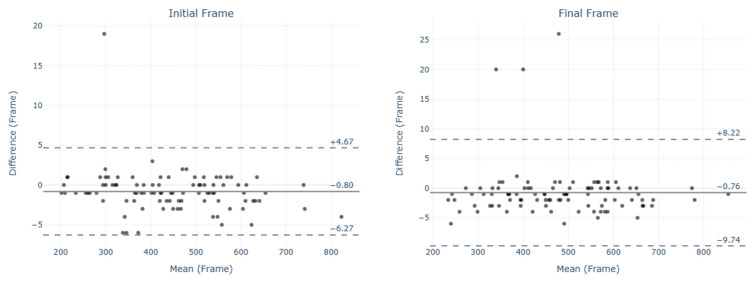
Bland–Altman plots for initial frame (IF) and final frame (FF) in the Vcut test. The solid line indicates the mean difference (bias), and dashed lines represent the limits of agreement (LoAs).

**Figure 6 sensors-25-00942-f006:**
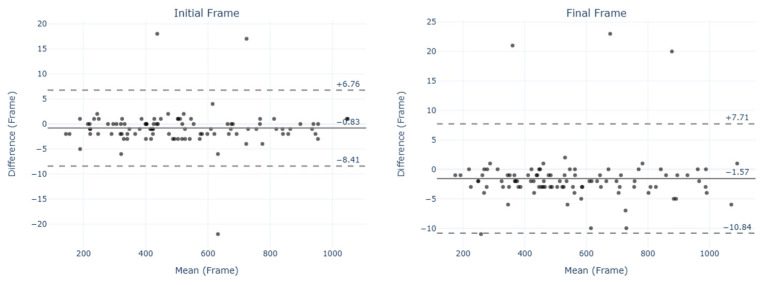
Bland–Altman plots for initial frame (IF) and final frame (FF) in the VcutBk test. The solid line indicates the mean difference (bias), and dashed lines represent the limits of agreement (LoAs).

**Figure 7 sensors-25-00942-f007:**
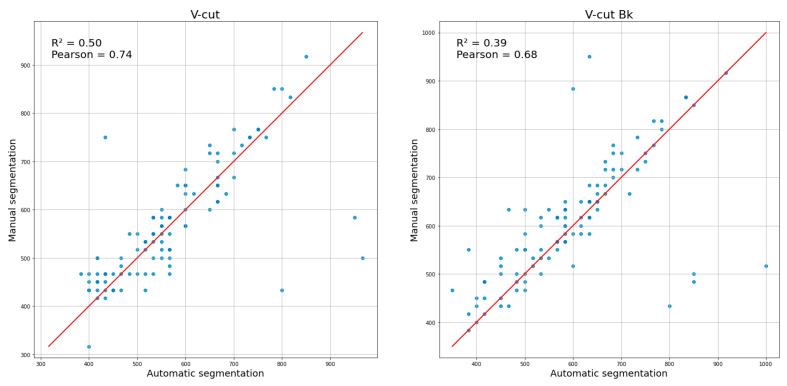
Correlation between manual and automatic segmentation for two conditions: V-cut (**left**) and V-cut Bk (**right**). The scatter plots analyze the data of total cutting time (in milliseconds), illustrating the relationship between the two segmentation methods. The coefficient of determination (R^2^) and Pearson correlation coefficient are reported in each panel. The red lines represent the bisector (y = x).

**Figure 8 sensors-25-00942-f008:**
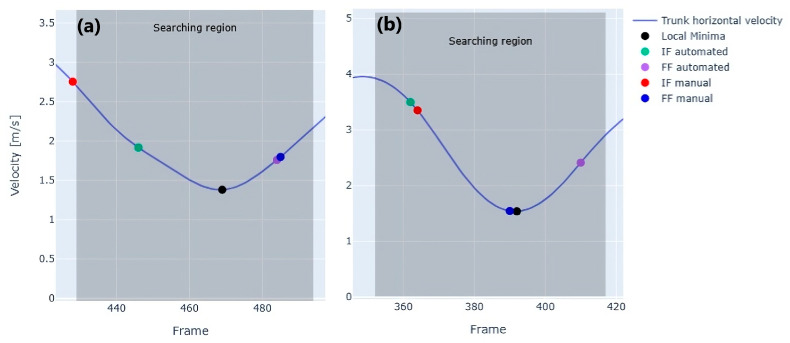
Examples of misdetection during COD events. (**a**) Error in detecting initial foot contact (IF), with the manually identified frame outside the search region. (**b**) Error in detecting final foot contact (FF), where the local minimum occurred after the actual event.

**Table 1 sensors-25-00942-t001:** Sample characteristics.

Team Category	Gender (*n*°)		Age (Mean ± SD)		Leg Dominance	
	**M**	**F**	**M**	**F**	**M**	**F**
Under 13	11	10	12.3 ± 0.2	12.1 ± 0.3	11 R; 0 L	9 R; 1 L
Under 17	10	12	16.3 ± 0.3	16.0 ± 0.3	7 R; 3 L	10 R; 2 L
Under 18	12	7	16.8 ± 0.6	16.8 ± 0.6	10 R; 2 L	5 R; 2 L

**Table 2 sensors-25-00942-t002:** Description of differences between methods, divided by test and frame identified in training and test datasets.

		Training Dataset			Test Dataset		
Test	Frame	Median	Mode	IQR	Median	Mode	IQR
V-cut	IF	−1	−1	2	−1	−1	2
	FF	−1	−1	2	−1	0	3
V-cut Bk	IF	−1	−1	2	−1	0	2
	FF	−1	−1	2	−2	−3	2

Note: IF = initial frame, FF = final frame.

**Table 3 sensors-25-00942-t003:** Summary of the automated method’s results of training and test datasets.

Training Dataset						
Test	Frame	|E|	s(E)	AP_2_	AP_3_	AP_4_
V-cut	IF	1.2	1.5	89.1	94.7	97.2
	FF	1.8	1.9	78.1	88.6	93.7
V-cut BK	IF	1.5	1.9	83.2	91.2	93.9
	FF	1.8	1.9	76.1	88.6	94.2
**Test dataset**						
**Test**	**Frame**	**|E|**	**s(E)**	**AP_2_**	**AP_3_**	**AP_4_**
V-cut	IF	1.7	2.7	80.2	89.6	95.8
	FF	2.4	4.4	69.8	92.7	94.8
V-cut BK	IF	2	3.8	78.8	90.9	93.9
	FF	2.9	4.6	55.6	79.8	86.9

Note: |E| = average mean error, s(E) = error standard deviation, AP = average precision. The bold formatting is used to differentiate the two datasets.

## Data Availability

The data are available upon reasonable request.

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
