# Peer review of "Automated Detection of Change of Direction in Basketball Players Using Xsens Motion Tracking"

_sensors, 2025, doi:10.3390/s25030942_

Round 1

Reviewer 1 Report

Comments and Suggestions for Authors

This is a manuscript comparing automated detection of change of direction to manual detection of change of direction in in basketball. The authors justify why automatic detection would be beneficial, but they do not adequately justify their specific approach to automation, nor do they describe the algorithm development in a way that others could reproduce. 

Key words should not appear in the title

The abstract leaves the reader wondering why change of direction is important.

Introduction: Other sports employ automatic segmentation and/or identification of events in kinematic data. For example, running analysis is automatically segmented by footstrike and toe off. Baseball pitching is automatically segmented with maximum knee height, foot contact, maximum shoulder external rotation, ball release, and maximum shoulder internal rotation. In these cases, event identification is typically identified as a maximum or minimum kinematic variable or a change in body segment velocity. It would be beneficial to introduce the typical ways kinematic data are automatically segmented and why this approach is not appropriate for your data/movement of interest. As is, the reader may wonder why you can’t just segment the data based on the trunk velocity minimum or even a time bracket around trunk velocity minimum.

Methods: An image of the experimental set up, with a dotted line marking the subjects path of motion would help the reader visualize the V-cut test.

In order for this work to be valuable to anyone else, the ‘algorithm’ needs more description of the development. The authors describe the variables used, but do not describe how the algorithm was developed. Given the data separation into training and test sets, I assume it was some type of neural network or machine learning process, but this is not described anywhere.

The variables used for algorithm development/segmentation are contained in this MVNX file that is specific to the Xsens motion system. Therefore this algorithm and automation process is not applicable to any other motion data. This should be addressed in the limitations.

Author Response

Comments 1:

This is a manuscript comparing automated detection of change of direction to manual detection of change of direction in in basketball. The authors justify why automatic detection would be beneficial, but they do not adequately justify their specific approach to automation, nor do they describe the algorithm development in a way that others could reproduce.

Response 1:

Thank you for your valuable comment. We recognize the importance of providing a more comprehensive justification for our approach to automation, as well as a detailed description of the algorithm to ensure reproducibility. In accordance with your comment, we have elaborated on the specific methodology underlying our approach and included a step-by-step workflow to clarify the algorithm development and implementation. This additional information is now provided in the Methods section.

Comments 2:

Key words should not appear in the title

Response 2:

Thank you for your observation. We acknowledge that the keywords should not appear within the list of keywords. In response, we have revised the keyword section to ensure it does not include terms already used in the title. We changed “change of direction” with “COD”.

Comments 3:

The abstract leaves the reader wondering why change of direction is important.

Response 3:

Thank you for pointing this out. We have revised the abstract to include a brief explanation of why COD is critical in basketball, highlighting its impact on performance and injury prevention. We added the following part to the abstract: “Changes of direction (COD), frequently executed during basketball games at cutting angles around 135° (internal angle of 45°), are essential for agility and high-level performance. Moreover, mastering effective COD mechanics is associated with a lower risk of injuries and enhanced long-term athletic success.”

Comments 4:

Introduction: Other sports employ automatic segmentation and/or identification of events in kinematic data. For example, running analysis is automatically segmented by footstrike and toe off. Baseball pitching is automatically segmented with maximum knee height, foot contact, maximum shoulder external rotation, ball release, and maximum shoulder internal rotation. In these cases, event identification is typically identified as a maximum or minimum kinematic variable or a change in body segment velocity. It would be beneficial to introduce the typical ways kinematic data are automatically segmented and why this approach is not appropriate for your data/movement of interest. As is, the reader may wonder why you can’t just segment the data based on the trunk velocity minimum or even a time bracket around trunk velocity minimum.

Response 4:

Thank you for your insightful comment. Our method combines well-established techniques, such as toe-off and heel-strike detection for segmenting strides using kinematic parameters, with trunk velocity analysis adapted from previous studies. While these methods have been applied individually in other contexts, their combined application to basketball COD movements, particularly at this specific angle, is novel.

We have revised the Introduction to better explain this integration and highlight its originality: “Several sports adopt automatic segmentation and event identification in kinematic data. In running, for example, foot contact events are commonly detected using maxima/minima of kinematic variables [26,27]. However, these methods may not capture the specific phases of basketball COD movements. A minimum trunk velocity or a fixed time window around it does not necessarily align with key foot-contact events in basketball, nor does it adequately account for the biomechanical demands of sudden deceleration.

Although these methods are well-documented in other sports, they have not been applied to basketball COD at a 135° angle. A segmentation strategy that adapts foot contact detection approaches from running [26,27] and combining them with horizontal velocity analysis [5,28] could address the unique demands of sudden decelerations in basketball. This approach helps fill a gap in the existing literature, as the standard methods used for linear activities often do not fully capture the complexity of basketball manoeuvres”.

Comments 5: 

Methods: An image of the experimental set up, with a dotted line marking the subjects path of motion would help the reader visualize the V-cut test.

Response 5:

Thank you for this valuable suggestion. We have added a figure (Fig. 1) to the “Experimental setup” section to provide a visual representation of the setup and the path taken by the participants during the test.

Comments 6: 

In order for this work to be valuable to anyone else, the ‘algorithm’ needs more description of the development. The authors describe the variables used, but do not describe how the algorithm was developed. Given the data separation into training and test sets, I assume it was some type of neural network or machine learning process, but this is not described anywhere.

Response 6:

Thank you for highlighting the need for more detailed information about the algorithm's development. To address this, we have clarified that the algorithm used in this study is a heuristic rule-based approach, not a neural network or machine learning method. We have added a comprehensive explanation of the algorithm's design process (Fig. 3), including the rationale for selecting specific parameters, the steps taken for optimization, and the workflow followed for its implementation. Additionally, we have clarified the rationale for dividing the data into training and test sets, which was implemented to minimize overfitting and ensure the robustness of the algorithm. We have expanded the Methods section to include specific subsections—2.6.2 Automatic Segmentation and 2.7 Model Training and Test—which describe the development process in detail. Additionally, the entire Methods section has been reorganized to improve the clarity and comprehensibility of the procedure.

Comments 7: 

The variables used for algorithm development/segmentation are contained in this MVNX file that is specific to the Xsens motion system. Therefore this algorithm and automation process is not applicable to any other motion data. This should be addressed in the limitations.

Response 7:

Thank you for your observation. We recognize the current dependency of our algorithm on the MVNX file format and specifically its reliance on the foot contact data provided directly by this format. According to your comment, we added this in the Limitations section as follows: “the algorithm currently relies on the MVNX file format of the Xsens motion system, particularly for its direct provision of foot contact events (e.g., toe-off and heel-strike). This limits the algorithm's direct applicability to other motion capture systems. However, the algorithm could be adapted by importing motion data from Excel/CSV files and utilizing alternative methods to identify foot contact and trunk’s horizontal velocity events through kinematic analyses (e.g sacrum accelerations [39] or foot accelerations [40]). These adaptations would enable broader applicability while preserving the algo-rithm's fundamental structure and capabilities.”

Reviewer 2 Report

Comments and Suggestions for Authors

The authors are requested to briefly describe in the abstract section what model was used for the detection of COD events.

1.Introduction:

Major:

[Comment 1] Authors are kindly requested to provide a concise explanation of the study’s purpose and hypothesis in the introduction section.

Minor:

[Comment 1] Line37-38: Please provide citations of relevant references that supports the author's perspective.

2. Materials and Methods

Minor:

[Comment 1] “Participants”: Regarding the sample size of subjects, the reviewer would like the authors to be realistic that the above sample size can satisfy this study by using scientific sample statistical methods. In addition, what are the recruitment requirements, exclusion criteria for participants in this experiment? In addition, were all participants on the right dominant leg?

[Comment 2] Line 103-104: The purpose of testing with and without a ball is not clearly described in the introduction section. The authors are encouraged to include this information to provide clarity.

[Comment 3] Line 104-106: The authors should clarify why the study only tested subjects with a 45° change of direction (COD), despite mentioning in the introduction that a COD beyond 60° may cause joint damage. This discrepancy requires explanation to align the study's testing parameters with the introductory information.

[Comment 4] Line 150-151: The study should provide detailed information about the "automated algorithm" that is integral to the research. This includes specifying the type, framework, and principles of the algorithm. Additionally, it is crucial to clarify whether the algorithm was developed by the author's team or sourced from a commercial company. This explanation is essential for transparency and understanding of the study's methodology.

4. Discussion

Minor:

[Comment 1] Line239-241: It is a fact that the authors do not describe exactly what automated methods were used in this study.

5. Conclusions

Minor:

[Comment 1] Line317-318:What are the results on which this conclusion presented by the authors is based. Some recently study could be added in the discussion, such as: Effects of Fatigue in Lower Back Muscles on Basketball Jump Shots and Landings, Physical Activity and Health, 6(1), p. 273286.

[Comment 2] Line 319: The review did not seem to see any COD content in the results regarding 135°.

Comments on the Quality of English Language

 The English could be improved to more clearly express the research.

Author Response

1.Introduction:

Major:

Comments 1:

Authors are kindly requested to provide a concise explanation of the study’s purpose and hypothesis in the introduction section.

Response 1:

Thank you for your comment. We rephrased the aim as follows: “In light of these, the aim of this study was to evaluate the feasibility and accuracy of an automated algorithm for detecting COD movements in basketball and to compare its performance with manual detection methods. We hypothesize that the automated approach, which uses specific kinematic parameters, will demonstrate accuracy comparable to that of manual methods, while providing a more efficient and reproducible analysis.”

Minor:

Comments 2:

Line37-38: Please provide citations of relevant references that supports the author's perspective.

Response 2:

Thanks for the comment. We added some reference to support our perspective as suggested.

  1. Li RT, Kling SR, Salata MJ, Cupp SA, Sheehan J, Voos JE. Wearable Performance Devices in Sports Sports Health. 2016;8(1):74-78.
  2. Benages Pardo, L.; Buldain Perez, D.; Orrite Uruñuela, C. Detection of Tennis Activities with Wearable Sensors. Sensors 2019, 19, 5004.
  3. Preatoni, E.; Bergamini, E.; Fantozzi, S.; Giraud, L.I.; Orejel Bustos, A.S.; Vannozzi, G.; Camomilla, V. The Use of Wearable Sensors for Preventing, Assessing, and Informing Recovery from Sport-Related Musculoskeletal Injuries: A Systematic Scoping Review. Sensors 2022, 22, 3225.

  1. Materials and Methods

Minor:

Comments 3:

“Participants” : Regarding the sample size of subjects, the reviewer would like the authors to be realistic that the above sample size can satisfy this study by using scientific sample statistical methods. In addition, what are the recruitment requirements, exclusion criteria for participants in this experiment? In addition, were all participants on the right dominant leg?

Response 3:

Thank you for your valuable comment. Regarding the sample size, we did not perform a formal statistical power calculation due to the exploratory nature of this study. However, the study was conducted on a heterogeneous and balanced sample, ensuring diversity in terms of sex, age, and test type. Furthermore, the training and test datasets were balanced to maintain the same proportion of males and females across age groups and testing conditions (Appendix Fig. S2). Additionally, the sample size was comparable to or larger than those in similar studies (Apte, S., Karami, H., Vallat, C. et al. In-field assessment of change-of-direction ability with a single wearable sensor. Sci Rep 13, 4518 (2023). https://doi.org/10.1038/s41598-023-30773-y).

We have clarified the recruitment criteria and exclusion criteria for participants in the manuscript. “The inclusion criteria required active involvement in basketball training, either at a competitive or recreational level, no history of lower-limb injuries or surgeries within the past six months, and a willingness to participate in all testing sessions. The exclusion criteria included any current injuries or conditions that could affect performance during the agility tests.”. 

Moreover, we added the information about the leg dominance in Table 1 and the following sentence: "The V-cut test design involves changes of direction to both the right and left sides, ensuring that performance evaluation is independent of participants’ leg dominance."

Comments 4:

Line 103-104: The purpose of testing with and without a ball is not clearly described in the introduction section. The authors are encouraged to include this information to provide clarity.

Response 4:

Thank you for your comment. The purpose of testing both with and without the ball was to evaluate the change of direction detection algorithm's applicability in both general athletic and sport-specific tasks, such as dribbling while handling a ball. This approach aimed to allow further analyses in scenarios closely resembling real match conditions. In order to clarify this purpose, we added the following part in Method section: “This dual approach aimed to evaluate the algorithm's performance in both general athletic and sport-specific tasks, such as dribbling while handling a ball in order to closely simulate real match conditions

Comments 5:

Line 104-106: The authors should clarify why the study only tested subjects with a 45° change of direction (COD), despite mentioning in the introduction that a COD beyond 60° may cause joint damage. This discrepancy requires explanation to align the study's testing parameters with the introductory information.

Response 5:

Thank you for bringing this point to our attention. The angles mentioned in the manuscript may have caused some confusion. The V-cut test evaluates changes of direction with an internal angle of 45°, which corresponds to an external angle of 135° from the initial running direction. These angles are effectively equivalent and describe the same movement pattern. We have revised the manuscript to ensure consistency in the definition and explanation of these angles throughout the text. We provided a more detailed explanation in Experimental setup subsection: “The V-cut test consisted of a 25-m sprint with four COD. Cones were arranged in a “V” shape, ensuring players crossed a marked line at each turn (to anticipate sidestep cutting movements) and requiring an actual COD at a 135° angle with respect to the athlete’s original running direction (corresponding to an internal angle of 45°). Each segment of the run measured 5 m in length.”

Comments 6:

Line 150-151: The study should provide detailed information about the "automated algorithm" that is integral to the research. This includes specifying the type, framework, and principles of the algorithm. Additionally, it is crucial to clarify whether the algorithm was developed by the author's team or sourced from a commercial company. This explanation is essential for transparency and understanding of the study's methodology.

Response 6:

Thank you for your comment. According to the reviewer, we have expanded the Methods section to describe the type, framework, and principles underlying the algorithm. We have added a comprehensive explanation of the algorithm's design process (Fig. 3), including the rationale for selecting specific parameters, the steps taken for optimization, and the workflow followed for its implementation. To address your concerns, we have expanded the Methods section to include specific subsection—2.6.2 Automatic Segmentation —which describe the development process in detail. Additionally, the entire Methods section has been reorganized to improve the clarity and comprehensibility of the procedure. 

  1. Discussion

Minor:

Comments 7:

Line239-241: It is a fact that the authors do not describe exactly what automated methods were used in this study.

Response 7:

Thank you for pointing this out. As mentioned in our response to your previous comment (Comment 6), we have clarified the automated methods used in this study. Specifically, we have detailed the development of the heuristic rule-based algorithm in Subsection 2.6.2 Automatic Segmentation of the revised Methods section. This includes the algorithm's framework, principles, and workflow, as well as the rationale for key design choices such as filtering and parameter selection.

We hope that this expanded explanation addresses your concern and provides the necessary clarity on the automated methods utilized in our study.

  1. Conclusions

Minor:

Comments 8:

Line317-318:What are the results on which this conclusion presented by the authors is based. Some recently study could be added in the discussion, such as: ‘Effects of Fatigue in Lower Back Muscles on Basketball Jump Shots and Landings’, Physical Activity and Health, 6(1), p. 273–286.

Response 8:

Thank you for your comment. We have reviewed the Conclusion section in order to explicitly highlight our results. “The automated COD detection system proposed in this study, based on trunk horizontal velocity and foot contact extracted from MVNX files, demonstrated high accuracy and precision in identifying COD events during a V-cut test, closely matching the performance of manual segmentation. This segmentation enables the identification of multiple COD movements at a 135° angle allowing for further kinematic analysis focused on this critical time period. This method holds significant potential as an effective tool for COD performance evaluation.”

Comments 9:

Line 319: The review did not seem to see any COD content in the results regarding 135°.

Response 9:

Thank you for your comment. As clarified in a previous response, the 135° angle refers to the external angle from the initial running direction, which is equivalent to a 45° internal angle used to describe the COD movements in this study. This reflects the same biomechanical action but from a different perspective. We have revised the manuscript to ensure the definition and use of these angles are consistent and clear throughout the text.

Round 2

Reviewer 1 Report

Comments and Suggestions for Authors

Thank you for revising according to reviewers comments. This is a much improved manuscript suitable for publication.